# Bovine Colostrum Applications in Sick and Healthy People: A Systematic Review

**DOI:** 10.3390/nu13072194

**Published:** 2021-06-25

**Authors:** Monica Guberti, Stefano Botti, Maria Teresa Capuzzo, Sara Nardozi, Andrea Fusco, Andrea Cera, Laura Dugo, Michela Piredda, Maria Grazia De Marinis

**Affiliations:** 1Research and EBP Unit, Health Professions Department, Azienda USL-IRCCS di Reggio Emilia, Via Amendola, 2-42122 Reggio Emilia, Italy; monica.guberti@ausl.re.it (M.G.); andrea.fusco@ausl.re.it (A.F.); 2Department of Biomedicine and Prevention, University of Rome Tor Vergata, Via di Montpellier, 1-00133 Rome, Italy; mariateresa.capuzzo@libero.it (M.T.C.); saranardozi95@gmail.com (S.N.); 3Hematology Unit, Azienda USL-IRCCS di Reggio Emilia, Via Amendola, 2-42122 Reggio Emilia, Italy; 4ASST-Azienda Socio-Sanitaria Territoriale di Mantova, Strada Lago Paiolo, 10-46100 Mantova, Italy; andrea.cera@asst-mantova.it; 5Unit of Food Science and Nutrition, Department of Science and Technology for Humans and the Environment, University Campus Bio-Medico of Rome, Via Alvaro del Portillo, 21-00128 Rome, Italy; l.dugo@unicampus.it; 6Research Unit Nursing Science, University Campus Bio-Medico of Rome, Via Alvaro del Portillo, 21-00128 Rome, Italy; m.piredda@unicampus.it (M.P.); m.demarinis@unicampus.it (M.G.D.M.)

**Keywords:** bovine colostrum, whey, milk, food supplementation, health improvement, physical performance, immune system, systematic review

## Abstract

Colostrum is the first secretion of mammalian glands during the early period after birth giving. Its components are biologically active and have beneficial effects on new-born growth and well-being. Bovine colostrum has the highest concentration of these substances and its supplementation or application may provide health benefits. This systematic review was conducted to update current knowledge on bovine colostrum effects including all administration routes on healthy and sick subjects. Full texts or abstracts of twenty-eight papers as reports of systematic reviews, randomized controlled trials, observational studies and case series were included after searches in Medline, Embase, Cochrane Library and Cinahl databases. The full texts of selected studies were assessed for quality using validated tools and their results were summarized in different categories. Studies were highly heterogeneous as regards to population, intervention, outcome and risk of bias. Bovine colostrum topical application was shown effective on vaginal dryness related symptoms limitation. Its use as food supplement showed interesting effects preventing upper respiratory illness in sportsmen, modulating immune system response and reducing intestinal permeability in healthy and sick subjects. Conflicting results were provided in pediatric population and little evidence is available on its use with older adults. Further studies are mandatory to better understand all factors influencing its activity.

## 1. Introduction

Colostrum is the secretion produced by the mammary gland immediately following parturition; it provides to infants sustenance, enhances their protection against pathogens, ensures immune system development, and provides to growth, maturation, and repair of several tissues [1]. The composition and physical properties of colostrum are highly variable due to a number of factors, including individuality, breed, parity, pre-partum nutrition, length of the dry period of cows and time post-partum [2,3,4,5,6]. It differs from milk as it contains less lactose and more fat, protein, peptides, non-protein nitrogen, ash, vitamins and minerals, hormones, growth factors, cytokines and nucleotides. The composition of both human and Bovine Colostrum (BC) have been largely studied highlighting higher concentrations than mature milk of a wide variety of biologically active substances [7]. These retain their activity while passing the gastrointestinal tract, carrying out beneficial effect on the intestinal functions [8,9,10] mainly attributable to immune-modulatory [11,12], antimicrobial [8,13] and anti-inflammatory [12] activities. Immunoglobulin antibodies are the main immune components of the acquired immune system present in colostrum and milk. Colostrum is characterized by its very high concentration of immunoglobulin G (IgG), which is of particular importance as it confers passive immunity to the neonate immediately following parturition [6,8,14,15]. The most abundant immunoglobulin class in bovine milk and colostrum is IgG1 [16]. In contrast, IgA and IgM are present at much reduced concentrations in bovine colostrum and milk [17]. Lactoferrin and lactoperoxidase have significant antimicrobial effects [18] and other components such as cytokines and interleukins are involved in inflammatory regulation processes and contribute to infection control also [12]. The highly glycosylated polypeptide secretory component [19] is another immune factor present in colostrum and milk that interacts with the adaptive immune system. Secretory component derived from a portion of the IgA receptor not only enhances IgA functionality when it is attached to IgA [20] but may have direct protective properties itself. Lysozyme is a lytic enzyme that plays a role in the innate immune system by attacking peptidoglycan cell constituents found primarily in gram-positive bacteria, leading to bacterial lysis [18,21]. Bovine colostrum has shown an emerging role as a food supplement due to its healing properties targeted to boost the immune systems in both healthy and chronically ill patients [1,18,22]. Colostrum has been known for centuries for its health benefits [23]. Literature also showed that the active components in BC were 100 to 1000-fold more concentrated than in human one. This means that even human infants can rely on cow or buffalo colostrum to gain health benefits [22,24]. Topical application of the colostrum constituents has proven promising for open wound healing [25]. It has been suggested that nucleotides, epidermal growth factor (EGF), transforming growth factor (TGF) and insulin like growth factor-1 (IGF-1) promote cellular and skin growth and also help in repairing DNA and RNA damage [6]. A patent has also been granted for the use of a colostrum fraction to accelerate wound healing [26]. For all these reasons, BC has found its application in the prevention and treatment of several inflammatory diseases, such as those affecting the gastrointestinal and respiratory systems of adults and children [27]. In addition, bovine colostrum derivatives have been used for the treatment of rheumatoid arthritis. Studies have shown benefits of BC supplementation in healthy populations such as children, adolescents [13] and sporting individuals, more precisely because it can boost their immune, digestive, and hormonal systems, as well as it may improve their physical performances [11]. The aim of this literature review is to assess the evidence supporting the bovine colostrum administration in clinical and not clinical settings in order to provide additional knowledge on its use and updating results of the last review on this topic [21].

## 2. Materials and Methods

### 2.1. Search Strategy

This review updated the last one published in 2014 [21]. The PICO method (P = Population, I = Intervention, C = Control, O = Outcome) was used to make the research question. Healthy and not-healthy subjects were included in the Population frame; Interventions were administration of BC (any formulation) with the aim to improve physical performances or health; all Control formulae including placebo and no-treatment were considered while the Outcomes were undefined. A literature search was conducted within the following databases: Medline, Embase, Cochrane Library and Cinahl. The following search strategy was used in Medline and adapted to the other databases: (“Colostrum” [Mesh] OR colostrum OR colostrums) AND (bovine OR cow OR cows OR cattle), records were filtered by time frame (from 1 January 2013 to 31 December 2020), language (English) and involved subjects (Human). Reference lists of included papers were manually checked in order to find further records. Full texts papers were considered in the analysis while abstracts were discussed separately.

### 2.2. Inclusion and Exclusion Criteria

Any population (adult, pediatric, aged, sick and healthy subjects), administration route (enteral, topical, etc.), endpoints and outcomes were considered to find results on BC use. Systematic reviews (SR), randomized controlled trials (RCTs), other clinical trials (CTs), observational studies (OS) and case series (CSs) were included. Full text papers or abstracts were considered as well as studies, investigating BC effects either as dietary supplement or as topical application. Records concerning hyper-immune colostrum, which are produced by administering polyvalent bacterial vaccines to cows with the aim of stimulating the production of higher levels of total and antigen-specific immunoglobulins [28] were excluded as well as those exploring the effects of composite products where colostrum was mixed with other bioactive substances. Any type of change, measurable in terms of health condition or physical performance, was considered as an eligible outcome for this review. The PRISMA Statement [29] was followed for results selection and reporting.

### 2.3. Study Selection and Quality Assessment

Study selection was performed by removing duplicates and non-relevant records, or protocols, first, on title and abstract basis and, second, after full-text reading (Figure 1). The quality assessment of the studies included was conducted with the aid of validated tools. The Effective Public Health Practice Project (EPHPP) [30] was used for clinical trials, randomized clinical trials (RCTs) and observational studies, while A Measurement Tool to Assess Systematic Reviews—2 (AMSTAR-2) [31] was used in order to assess reviews. The EPHPP is widely used in evaluating RCTs and clinical trials, with an excellent degree of inter-rating reliability [30]. This tool consists of six domains (*selection bias, study design, confounders, blinding, data collection methods, withdrawals and drop-outs*), whose sum of scores constitutes the overall quality rating, where (1) “strong” includes papers without weak sub-scale grading, (2) “moderate” includes those with at least one weak grading, and (3) “weak” includes two or more weak rating of the sub-scales. The AMSTAR-2 tool consists of 16 items with an overall rating based on weaknesses in critical domains. The overall rating is classified as “high” (No or one non-critical weakness), “moderate” (More than one non-critical weakness) “low” (One critical flaw with or without non-critical weaknesses) and “critically low” (More than one critical flaw with or without non-critical weaknesses). Quality assessment of the included papers with EPHPP and AMSTAR-2 was performed by two independent reviewers, who met to discuss and solve any discrepancies in the study evaluation or results interpretation. With the aim to intercept minimal differences in studies quality and to better define level of evidence, two additional tools were applied. The risk of bias of the included RCTs was assessed by two different independent reviewers using the Cochrane Collaboration Risk of Bias Tool (CCRBT) including the following seven domains: *random sequence generation; allocation concealment; blinding of participants and personnel; blinding of outcome assessment; incomplete outcome data; selective reporting and other biases*. The CCRBT is different from EPHPP as, for example, outcome reporting is evaluated more in detail than in the EPHPP, which gives no specific evaluation on absence from reporting bias [30]. Then, the level of evidence was assessed by two independent reviewers using the 2011 Oxford Centre for Evidence Based Medicine level of evidence method (OCEBM), which was developed by an international group and took into account feedback from clinicians, patients, and researchers. It allows to rapidly find the likely best evidence encouraging clinicians, researchers and patients to autonomously assess evidence [32] (Table 1).

## 3. Results

### 3.1. Paper Selection and Categorization

The literature searching identified 423 records: 137 from Medline, 165 from Embase, 77 from the Cochrane Library and 44 from Cinahl. After removal of 155 duplicates, 268 articles were considered appropriate in accordance to the screening of title, abstract and full-text reading. Finally, 28 articles were included in the review. Figure 1 represents the Flow Chart of the study selection process. Twenty-four out the 28 studies included the full texts [33,34,35,36,37,38,39,40,41,42,43,44,45,46,47,48,49,50,51,52,53,54,55], while 4 were available as abstracts only [56,57,58,59]. The majority of these articles involved participants from Europe (*n* = 14; 52%), 6 (22%) from Asia, 3 (11%) from Africa 1 (4%) from America and 3 (11%) from Oceania. Twenty-four studies involving 1057 participants were RCTs (4 available as abstracts only), 3 were observational studies with 588 patients and 1 was a systematic review. Therefore, the papers were classified by publication type (full texts or abstracts), BC administration route (systemic oral or topical) and study population. BC topical application in uro-gynecology [33,34,35] (Table 2), BC as dietary supplement in the sporting population [36,37,38,39,40,41,42,43,44,45,46] (Table 3), BC as dietary supplement in the pediatrics and preterm infants [47,48,49,50,51,52,53] (Table 4), BC as dietary supplement in the elderly [54] (Table 5), BC administration in critically ill patients [55] (Table 6), abstracts [56,57,58,59] (Table 7) and then a systematic review on BC clinical applications [21] (Table 8). In Table 9 the risk of bias of the included (full texts) RCTs was reported according to the CCRBT method.

### 3.2. Studies Heterogeneity

Three studies (1 RCT, 2 OS) were on topical applications of BC-based creams on various vaginal conditions and recruiting only female participants with a mean age ranged from 27 to 61 years [33,34,35]. Twenty-four papers (23 RCTs, 1 OS) assessed the efficacy of BC as dietary supplement in various populations [36,37,38,39,40,41,42,43,44,45,46,47,48,49,50,51,52,53,54,55,56,57,58,59], including diseased ones [47,48,49,50,51,52,53,55,56,57,58,59] and healthy people [36,37,38,39,40,41,42,43,44,45,46,54]. Of these papers, 1 included 3 RCTs (please, note that it is considered as one study in this review) [39], 11 recruited only sporting males with a mean age ranged from 21 to 51 years [36,37,38,39,40,41,42,43,44,45,46], 11 involved pediatric subjects affected by various clinical conditions [47,48,49,50,51,52,53,56,57,58,59], 1 was on critically ill patients and 1 on healthy elderlies. The participants involved in the 23 full texts of primary studies included received 11 different commercially available BC formulas, while one study experimented the effects of “fresh” BC [47]. In addition, BC was administered at different dosages ranging from 1 to 60 g/day in adult setting (13 RCTs) [36,37,38,39,40,41,42,43,44,45,46,54,55] and from 0.014 g/day to 4.5 g/kg/day in the 7 pediatric ones (7 studies) [47,48,49,50,51,52,53]. Colostum and placebos were administered majorly as beverages supplementing meals in clinical settings (12 studies) [47,48,49,50,51,52,53,55,56,57,58,59] and supporting physical performances or specific training programs in healthy people (12 RCTs). Refs. [36,37,38,39,40,41,42,43,44,45,46,54] The control groups of the 19 included full texts RCTs received 9 different placebos including isoenergetic/isomacronutrient formulas, whey or mixed milk matrix, whey protein concentrated formulas, corn flour, maltodextrin. Limited information on BC and placebo products were available by the 4 RCTs (all involving pediatric subjects) available as abstracts [56,57,58,59]. Surrogate outcomes were used in 15 studies [37,38,39,40,41,42,43,44,45,46,47,50,51,54,55] while 8 [33,34,35,36,48,49,52,53] assessed morbidity outcomes. The 4 abstracts measured primary outcomes. Refs. [56,57,58,59] As explained above, the included papers were very heterogeneous precluding any meta-analysis in this review.

### 3.3. BC Topical Applications in Uro-Gynecology Setting

BC was used as component of topical ointments in the treatment of different vaginal conditions (Table 2). Nappi et al. [33] studied the efficacy of a BC-based gel cream (Monurelle Biogel^®^) on Vaginal Dryness (VD) of 95 women randomized in two groups, patients in the Treatment Group (TG; *n* = 48) received BC gel during the intermenstrual period (23 days) versus no treatment in the Control Group (CG; *n* = 47). Clinical success was defined as the reduction of the vaginal discomfort of at least 1 point on a 5 points Verbal Rating Scale after the treatment period. Vaginal symptoms were significantly decreased in TG (92.7% vs. 63.6%; *p* = 0.002). Assessing vaginal health, significant improvements of the Vaginal Health Index (VHI) scores were obtained in TG than CG (4.4 ± 2.6 vs. 1.2 ± 1.9; *p* < 0.001), as well as the sexual function evaluated by the Female Sexual Function Index (FSFI) (*p* < 0.032). No differences were on Female Sexual Distress Scale-revised (FSDS-R) scores (*p* = 0.16). The same product was used daily for 12 weeks in a large retrospective cohort (*n* = 172) of women (mean age 61 years) with Vulvo-Vaginal Atrophy (VVA) [34]. In this study VHI mean scores were improved from 12.5 ± 3.67 at baseline to 19.3 ± 3.49 after treatment (*p* < 0.001), FSFI scores from 21.64 ± 2.99 to 28.16 ± 1.93 (*p* < 0.001) and FSDS scores decreased from 20.52 ± 5.90 to 8.15 ± 4.18 (*p* < 0.001). The effectiveness of BC-based vaginal tablets (Ginedie^®^) on Cervical Intraepithelial Neoplasia low grade lesions (CIN1) caused by the Human Papilloma Virus (HPV) was retrospectively assessed in a large group (*n* = 256) of patients (mean age 38 years) [35]. After 6 months of treatment (twice a week night time) 71% of cases were histologically negative while in 27% CIN1 histology was confirmed and in 2% the evolution to CIN2 was observed.

### 3.4. BC as Dietary Supplement in the Sporting Population

Eleven double blinded placebo controlled randomized trials reports were included in this category [36,37,38,39,40,41,42,43,44,45,46]. One of them included three RCTs [39] and 3 trials adopted a crossover design [39,43,45]. These trials recruited only male subjects (*n* = 289; range 10–57 participants; mean age range 22–51 years) at various level of training including regularly/recreationally trained people (7 RCTs) [36,37,38,39,43,44,45] and highly trained ones such as cyclists, soccer players and professional fighters (4 RCT) [40,41,42,46]. Desiccated BC and placebo formulations were administered to the participants within beverages. The dosages varied among studies from 1 to 60 g/day as well as the exposition duration was variable from 4.5 h to 12 weeks. The majority of control groups (*n* = 5) received isoenergetic and isomacronutrient placebo [36,37,38,39,43]; whey proteins (*n* = 4) [40,41,42,46] and corn flour (2) were also used. In 3 trials, BC and placebo were administered as dietary supplement [36,37,40] without other variables modification while in 8 studies were associated to various exercise programs. These programs differed between them for intensity and duration [38,39,41,42,43,44,45,46]. In one study BC was associated to desiccated banana [40] and in another one was administered as colostrum protein concentrate [46]. A wide heterogeneity was present regarding assessed outcomes. The efficacy of BC preventing Upper Respiratory Illnesses (URI) were evaluated by one trial [36] (53 participants; mean age 51 years) that reported significant difference between groups on URI incidence (0.4 ± 0.7 vs. 0.8 ± 0.7; *p* = 0.03) after 12 weeks of treatment and a significantly lower proportion of URI duration (days) (0.05 vs. 0.09; *p* < 0.001). One study including 3 RCTs reported findings on safety [39] measuring IGF-1 blood level variations after assumption of 40 g of BC during 4.5 h of moderate exercises (*n* = 16 participants), and then after 4 (*n* = 20) and 12 weeks (*n* = 57) of 20 g/day supplementation. No significant changes (per study group and per time) were reported. The gut permeability was assessed by 3 RCTs [40,43,45]. BC was more effective than placebo reducing sugar absorption (*p* = 0.01) and zonulin stool concentration (*p* = 0.03) after 20 days of supplementation in 16 athletic males during competitive period [40]. In a crossover trial (*n* = 20), Intestinal Fatty Acid Binding Protein (I-FABP) plasmatic level increasing were significantly less (*p* = 0.015 and *p* = 0.019 at the end and 1 h after exercises respectively) in treatment group of regularly exercised males undergoing study specific exercise program (heat condition) after 14 days of BC supplementation (20 g/day) [43]. The same outcome was evaluated in another RCT recruiting 57 trained and untrained subjects supported for 7 days with BC or placebo at 1.7 g/kg/day [45]. Immediately after the study exercise, I-FABP blood levels significantly increased than baseline in both groups (*p* < 0.001), but higher values were in trained group compared with untrained one (*p* = 0.006). Two RCTs by Jones et al. [37,38] (*n* = 34, *n* = 20, respectively) evaluated the effects of BC on immune response. BC supplementation (20 g/day) for 58 days blunted immune response attenuation due to physical activity and increasing immune sensitivity at 24 h (*p* < 0.001) and at 48 h (*p* = 0.023). No significant effects on in-vivo immune responsiveness, IGF-1 blood levels, immune cell counts [37]. Improved attenuation of the decline in formyl-Methionyl-Leucyl-Phenylalanine (fMLP) stimulated oxidative burst response in a model of exercise-induced immune dysfunction after 4 weeks of BC administration (20 g/day) was showed in a little placebo controlled trial (10 subjects per arm) (*p* < 0.05). No significant effects on other receptors or on mucosal barrier function were observed [38]. The effects of BC (3.2 g/day for 6 weeks) on muscular health and performance maintaining were studied in two trials [41,42] involving 44 soccer players (22 per RCT) undergone the Loughborough Intermittent Shuttle Test (LIST). The decline of rate of torque was reduced in both groups (intervention and placebo) without significant difference as well as no difference resulted by maximum isometric voluntary contraction, countermovement jump and perceived muscle soreness comparisons. Some beneficial effects were showed on faster recovery of squat jump performance and of some biochemical parameters. The effects of BC on blood brain barrier (BBB) permeability and cognitive function (CF) were assessed by one trial [44] recruiting 15 trained and untrained men in a crossover design study. BC and placebo were administered for 7 days at 1.7 g/kg/day before a 90 min exercise program in heat conditions. No significant results were obtained. Significant differences were showed in a pilot study by Shing et al. [46] on hormonal and autonomic response of cyclists assigned to BC (*n* = 4) supplementation (10 g/day over 8 weeks of colostrum protein concentrate) than placebo (whey protein concentrate) (*n* = 6). Table 3 summarize further information on this category.

### 3.5. BC as Dietary Supplement in Pediatrics and Preterm Infants

Six RCTs and 1 OS were in this category, all of them exploring the effects of BC administration as dietary supplement (Table 4) [47,48,49,50,51,52,53]. Five RCTs were double blinded placebo controlled [47,48,49,51,52] while 1 was controlled versus standard practice without blinding [50]. The studies recruited 548 patients (RCTs *n* = 388, OS *n* = 160), 290 males and 258 females including toddlers [47,53], newborns [48], infants and toddlers [49], preterm newborns [50], any children [51,52]. Seven different diseased populations were involved in the 7 studies: Short Bowel Syndrome (SBS) [47], Very Low Birth Weight (VLBW) newborns [48], acute diarrhea and vomit [49], preterm newborns [50], IgA deficient children [51], Acute Lymphoblastic Leukemia (ALL) [52], recurrent acute URI or diarrhea [53]. BC was administered within beverage (various powder formulas) in 5 studies [48,49,50,52,53], sucking tablets in 1 [51] and as “fresh product” in the study by Aunsholt et al. [47] It was administered at various dosages ranging from 14 mg to 4.5 g/kg/day and following different protocols (from 1 to 12 weeks, from 1 to 4 time per day) based on ageing, underlying diseases, nutritional needs and study protocols. Aunsholt et al. studied the supplementation with BC (versus milk mixed matrix) of 20% of the Basal Fluid Requirement (BFR) in children (*n* = 9) affected by SBS. A crossover designed RCT was performed. The lack of significant differences among groups on energy (*p* = 1) and wet weight absorption (*p* = 0.93) suggested that BC dietary supplementation did not improve intestinal function in these patients [47]. Balachandran et al. evaluated the efficacy of BC in a pilot study comparing BC (1.2–2.0 g/dose, 4 time/day for 21 days) and placebo in 86 VLBW neonates. No significant differences on incidence of Necrotizing Enterocolitis (NEC), sepsis and mortality were found between groups (*p* = 0.4, *p* = 0.4, *p* = 1 respectively). However, a trend toward a higher incidence of NEC in BC group was showed (9.3% vs. 2.1%, RR = 4.31, 95% CI = 0.42–105.82) [48]. An RCT recruited 160 children (6 months-2 years) with acute diarrhea [49] randomizing them to receive BC (3 g/day for 1 week) or placebo in addition to standard therapy. After 48 h, the BC group had a significantly lower frequency of vomiting, diarrhea, fever and Vesikari scoring compared with the placebo group (*p* < 0.001, *p* = 0.001, *p* < 0.001, *p* < 0.001, respectively). Significant results on number of children with diarrhea were reported after 72 h and 7 days (35% vs. 94%, *p* < 0.001; 0% vs. 12.5%, *p* = 0.001 respectively) suggesting beneficial effects of BC supplementation in this population. Tolerability and safety of BC (max 4.4 g/kg/day for 10–14 days) in addition to standard feeding (mother milk, donor milk or infant formula) versus standard feeding alone were studied on 40 preterm neonates [50]. A number of Adverse Events (AEs) without significant differences between groups were reported by this study, including 1 death (NEC not related to colostrum supplementation), 3 sepsis, 1 bronchopulmonary dysplasia, 2 retinopathies, 4 intraventricular hemorrhages. No significant differences in dietary intolerances were reported. No significant difference in salivary IgA secretion was reported in a RCT involving 31 IgA deficient patients who received BC 14 mg + lysozyme 2.2 mg sucking tablets 3 time per day for 1 week [51]. Rathe et al. in a multicenter RCT recruited 62 pediatric patients affected by ALL (age range 1–18 years) [52]. Patients were randomized to receive a BC or an isocaloric placebo (0.5–1 g/kg/day) during 4 weeks of chemotherapy treatment. Data on fever level and duration, bacteriemia episodes, mucositis severity and biochemical parameters were collected. Severity of oral mucositis (peaks) resulted significantly reduced in BC group (*p* = 0.02); however, BC did not show any effects on fever, infections and other outcomes. A large cohort prospective multicenter study by Saad et al. enrolled 160 children (81 males, 79 females) between 1 and 6 years with recurrent infectious URI or diarrhea [53]. The cohort was supplemented for 4 weeks with 3 g/day of BC. Reduction of infection episodes were observed at 2 and 6 months follow up (mean 8.6 ± 5.1 baseline vs. 5.5 ± 1.2 after 2 months; *p* < 0.001 vs. 5.7 ± 1.6 after 6 months; *p* < 0.001) and the reduction of hospitalization frequency was reported (*p* < 0.001). Further details on included study of this category were in Table 4.

### 3.6. BC as Dietary Supplement in Healthy Older Adults

Only one paper was found in this field. The RCT by Duff et al. compared the effects of BC versus whey protein complex (60 g/day for 8 weeks) randomizing 40 healthy elderlies (mean age 59 years, 15 males and 25 females) undergone a study specific exercise training program. Outcome such as body composition, strength, muscle thickness, bone resorption, cognitive function and biochemical exams were assessed in the trial. In the treatment group, the leg press strength was significantly higher (24 ± 29 kg, *p* < 0.01) while urinary N-telopeptides (Ntx), that is considered a marker of bone resorption, was significantly reduced (*p* < 0.05). No difference on other outcomes were found [54].

### 3.7. BC Administration in Critically Ill Patients

A double blinded placebo controlled randomized trial was identified that assessed colostrum intake in critically ill patients (*n* = 70, mean age 62 years). A dosage of 20 g/day of BC or placebo (isocaloric maltodextrin) were added to enteral feeding for 10 days [55]. Intestinal permeability was the primary end-point and it was assessed measuring plasmatic endotoxin and zonulin concentrations. Plasmatic endotoxin concentration and zonulin levels were significantly lower in BC group after 10 days of treatment (*p* < 0.05 and *p* < 0.001, respectively). In addition, in BC group the incidence of diarrhea was lower (*p* = 0.02).

### 3.8. Abstracts on BC Clinical Applications

Four of 28 papers included in this review were only available as abstracts. All of them [56,57,58,59] were placebo controlled RCTs and concerned the pediatric setting. They showed BC benefits increasing neutrophil account in LLA patients who underwent chemotherapy [56]; decreasing acute diarrhea in infants after 72 h [57], reducing enteric inflammation in exposed children [58] and reducing nasal congestion promoting pulmonary function in children affected by respiratory allergies [59]. The authors of this review thought it was important to include these abstracts because they might highlight new research options or confirm results of other studies. However, it was not possible to perform a quality assessment of these records due to poor available information.

### 3.9. Systematic Reviews on BC Clinical Applications

Table 8 summarize the results provided by the last systematic review on clinical applications of BC (enteral intake only).

## 4. Discussion

Three included studies considered different BC topical applications in uro-gynecology setting [33,34,35]. VD was often associated with a genitourinary syndrome, leading to sexual dysfunction and poor quality of life of affected women; the vaginal gel formulation (Monurelle Bio-gel^®^) used in two studies included in this review [33,34] showed characteristics similar to the physiological vaginal secretions creating a conducive to natural lubrication environment. Considering the high patients’ compliance and the lack of AEs reported by these studies, very encouraging results were found on vaginal and sexual health as well as urinary symptoms. Despite these findings were available by one RCT alone without blinding strategies, the reviewer considered robust its results attributing high level of evidence supporting the use of this formulae to reduce VD symptoms in young women (OCEBM and EPHPP level 1 evidence). In elderly women this approach seemed to be effective managing VD due to vaginal atrophic issues [34]. However, weak evidence (Table 2) may be provided in this population and these findings should be confirmed by well-designed RCTs. The immune-stimulatory and nutritive functions of topical application of BC could have a positive effect in the management of CIN1 lesions [35]. However, the level of evidence provided on this topic (OCEBM 4 and EPHPP 3) should be improved with randomized controlled studies. The humectant, moisturizing, re-epithelizing, antioxidant and immune-stimulant activities of BC were effective improving vaginal health and reducing sexual distress in women affected by VD conditions. Furthermore, the topical use of BC appeared safe and inexpensive. Further well-designed studies such as randomized controlled trials should be conducted to increase the level of evidence of the results on CIN1 and to explore the effects of BC topical applications in other conditions.

Bovine colostrum was used as dietary supplement in the sporting population. The studies included explored the benefits of BC as natural sporting nutritional supplement on subjects’ well-being and physical performances [36,37,38,39,40,41,42,43,44,45,46]. Jones et al. [36], used BC as a nutritional supplement to boost immunity and reduce the risk of URI in athletes. The study showed that BC supplementation limited salivary bacterial load and reduced URI more significantly than placebo supplements. Some selection bias-related concerns (Table 9) and the characteristics of the sample (only males) limited the level of evidence provided by this study. As reported by Rathe et al. in the systematic review included here [21], the relationship among physical exercise intensiveness and URI development depends on many factors and is not well recognized, yet. In agreement with the previous findings, not conclusive evidence were provided by our review due to study shortage, heterogeneity and risk of bias. Further well designed RCTs are warranted in this population. The safety of BC use during physical activity was explored. IGF-1 plasmatic levels did not increase during the short and long periods after the administration of standard doses of BC associated to training programs [39]. Intestinal permeability may change due to various factors such as inflammation or heat stress produced by intensive physical activity [38]. The safety and effectiveness of BC supplementation intake on athletes’ health were demonstrated in another placebo-controlled comparison where the reduction of the intestinal permeability was highlighted measuring zonulin stool concentration and sugar absorption [40]. However, some doubts on reporting bias and the very little sample size did not allow to provide strong evidence on these findings. Intestinal fatty acid binding protein (I-FABP) was considered a marker of gut heat-related permeability changes in subjects undergone intensive physical exercises [60]. The effectiveness of BC supplementation reducing its plasmatic level increasing after intensive activity was detected in a placebo controlled RCT included in this review [43]. However, despite there was a main effect in the treatment arm, this trial reported no statistically significant effect on bacterial plasmatic DNA concentration. These results were in contrast with the study by Morrison et al. [45] that showed how prolonged exertion in hot environment enough to produce heat stress increased gastrointestinal permeability independently by subjects’ level of training. I-FABP was higher in colostrum group and more in trained people than untrained ones in this study. Further studies are mandatory in order to provide further knowledge on factors influencing intestinal damage and its permeability changes during exercise with or without colostrum supplementation. Weak evidence of no effects were provided evaluating the efficacy of BC on blood brain barrier permeability and cognitive function [44]. Despite evidence of effects were showed on some outcomes, inconclusive findings were available on BC effects on immune system responsiveness during intensive exercises (neutrophil response and innate immunity) [37,38] as well as on BC effects on physical performances and muscle damaging [41,42] due to the lack of response on other outcomes considered in these studies, the large use of surrogate endpoints and their very small sample sizes. Interesting results was showed examining the effects of BC on participants’ hormonal profile that could have impact in recovery improvement and fatigue reduction; however, these results must be carefully considered due to very small sample size of the pilot study included [46]. An agreement would be needed in scientific community defining the endpoints to be included in well-designed randomized controlled trials to better explore the effects of BC in athlete populations.

The seven studies included in this review exploring the use of BC as dietary supplement in pediatric population and preterm infants were very heterogeneous in terms of the target population, outcomes, and study designs. Short Bowel Syndrome (SBS) causes nutrients and fluids malabsorption and may lead to severe consequences such as necrotizing enterocolitis, volvulus or gastroschisis [61]. The pilot RCT included in this review [47] hypothesized that the BC high concentration of peptide hormones, immunoglobulins, macro and micronutrients could implement intestinal absorption in SBS children. However, confirming Rathe and colleagues findings [21], evidence of no effects on intestinal absorption and metabolic balance of BC supplementation in this population was reported by our review. Although a protective effect of BC on gastrointestinal mucosa was suggested, there were evidence of no effects of BC supplementation on neutropenic fever, antibiotic drugs utilization and bacteriemia incidence in Acute Lymphoblastic Leukemia pediatric patients undergoing chemotherapy. A large cohort study by Saad et al. [53] showed that oral administered BC prophylaxis in children affected by recurrent Upper Respiratory Tract Infections (URTI) and diarrhea reduced both infection and hospitalization rates maintaining a protective effect in the long time period. The direction of these results was confirmed by two double-blind RCTs with placebo where BC was effective reducing severity of URTI affecting IgA deficient patients [51] and decreasing frequency and severity of both vomiting and diarrhea in infants with gastrointestinal infection diseases [49]. The authors hypothesized that the positive effect of BC may be traced to its high concentration of immunoglobulins (IgG, IgM, IgE, and IgD), lactoferrin and lysozyme. These findings should be still managed carefully in relation to patients’ clinical condition and ageing due to controversial results achieved on BC supplementation in frailest populations of this category (preterm and VLBW infants) [48,50]. In these two settings BC administration for mothers’ milk supplementation resulted a feasible option for protein intake enhancing without increasing feeding intolerance and without any side effects, apparently [50]. However, a number of AEs was reported in preterm infant population (equally distributed among groups) [50] and a trend of increased frequencies of necrotizing enterocolitis and sepsis were observed in treatment group [48]. These findings might due to the frailty of the studied populations (AEs) as well as to the production process of the colostrum-based product used [48]. It is known that industrial processed formula products may increase necrotizing enterocolitis and sepsis in this population [50]. Despite BC supplementation in pediatrics suggested protective effects on respiratory infectious diseases and other conditions related symptoms, its role preventing gastrointestinal issues or supporting preterm infants remain controversial. Further studies are needed in order to provide new evidence on BC administration as dietary supplement in pediatric populations (such as those with cancer) and to clarify the effects of its different preparations.

BC was used as dietary supplement in healthy older adults in a single double blind RCT, where BC intake outcomes were assessed during a specific resistance training and compared with those of whey proteins administered as placebo. BC was effective increasing legs strength (OCEBM level 2 and EPHPP level 1 evidence) and reducing bone resorption. No significant results were found on inflammation, upper body performances and body composition. These findings were interesting in order to promote further investigations on BC effects on bone health in this population [54].

The effects on gastrointestinal permeability of early enteral bovine colostrum supplementation versus placebo were investigated in intensive care unit (ICU) setting [55]. Significant reductions of plasmatic concentrations of both zonulin and endotoxin were found (high level of evidence), meaning safe effect reducing intestinal permeability in these patients. In addition, positive results were found on length of ICU stay and diarrhea. No effects were found on mortality, sepsis and other gastrointestinal outcomes.

Two systematic reviews were published in the time period considered. The review by Blair et al. [62] was aimed to evaluate the benefits of both milk and BC as dietary supplements in the healthy adult population ≥35 years old. However, this review included only one study on BC already selected in our review, then the review was excluded as not pertinent. The review by Rathe et al. [21] included 51 reports including RCTs, observational and case series studies published until March 2013. The review assessed the effects of BC enteral supplementation. Evidence of BC’s benefits was reported on patients’ intestinal tract protection from non-steroidal anti-inflammatory drugs (NSAID) induced injuries, and on bacterial translocation through gut barrier in patients undergoing abdominal surgery as well as on immuno-compromised patients’ (Human Immunodeficiency Virus) diarrhea control. In addition, inconclusive evidence were showed on growth or metabolic issues and idiopathic arthritis. We could not update these findings in our review due to the lack of further studies. Despite was clear that BC’s activity may triggers bowel immunological events that provides systemic ones, and its biological effects on intestinal permeability and upper respiratory tract health were explored suggesting a role of BC supplementation in URI prevention and intestinal infections treatment in both adult and pediatric populations; the interaction mechanisms of BC components with the immune system are largely underexplored. This suggests the need of pre-clinical (in-vitro and in-vivo) studies to better understand the process that influences its effects.

## 5. Conclusions

This review highlighted multiple clinical applications of BC and confirmed some general benefits on intestinal and respiratory recovery in absence of adverse effects. BC seemed to promote immune system enhancing and modulating local and systemic responses in various clinical and not clinical conditions. However, the studies’ heterogeneity regarding included populations, sample sizes, intervention and control protocols, and outcomes did not allow to perform meta-analyses. Moreover, the risk of biases and the large use of surrogate endpoints in the studies included did not consent to provide strong evidence on its use in any situation. Further well-designed studies are needed to support the administration of BC in adult, pediatric, clinical and not clinical settings. Pre-clinical studies should be performed to improve knowledge on BC effects.

## Figures and Tables

**Figure 1 nutrients-13-02194-f001:**
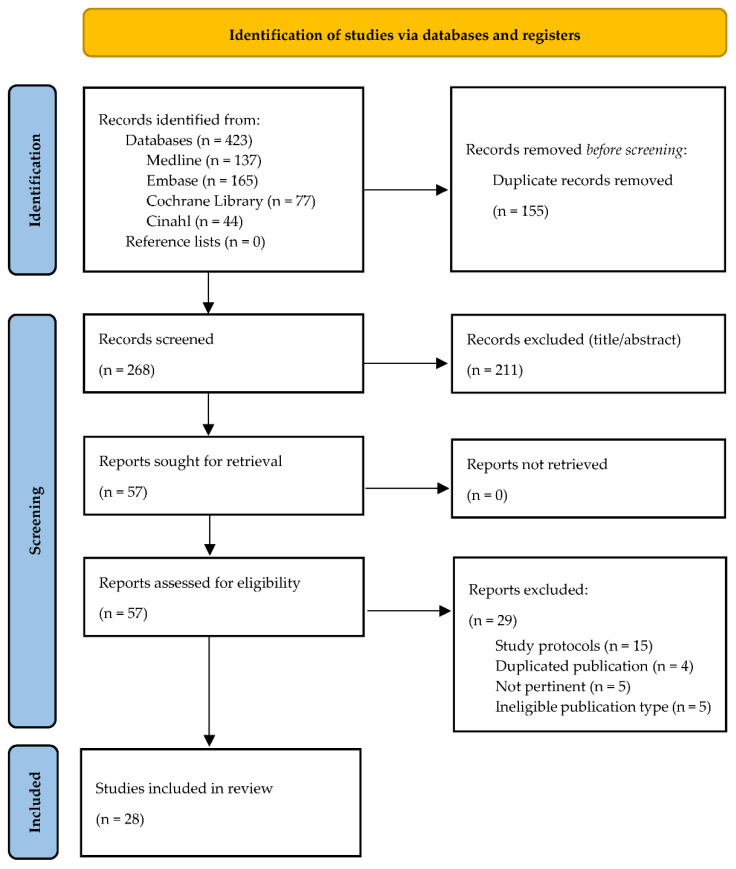
PRISMA flowchart of the study selection process.

**Table 1 nutrients-13-02194-t001:** OCEBM Level of Evidence 2011 [32].

Evidence Level (Treatment Benefits)	**Level 1 ***
Systematic reviewof randomized trials or n-of-1 trials

**Level 2 ***
Randomized trial or observational study with dramatic effect

**Level 3 ***
Non-randomized controlled cohort/follow-up study **

**Level 4 ***
Case-series, case control studies, or historically controlled studies **

**Level 5 ***
Mechanism-based reasoning

* Level may be graded down on the basis of study quality, imprecision, indirectness (study PICO does not match questions PICO), because of inconsistency between studies, or because the absolute effect size is very small; Level may be graded up if there is a large or very large effect size. ** As always, a systematic review is generally better than an individual study.

**Table 2 nutrients-13-02194-t002:** BC topical applications in uro-gynecology setting.

Authors	Study Design	Population Number Groups Gender Mean Age	Intervention Matrix	TG Size Dosage Frequency Duration	Control Matrix	CG Size Dosage Frequency Duration	Endpoints Data Collection Tools	Adverse Events	Results	OCEBM	EPHPP
Nappi, R. E., et al. (2016) [33]	RCT	Women with vaginal dryness. *n* = 95; >18 ys	Monurelle Biogel^®^	TG (*n* = 48) 5 mL, 1–2 time per day during intermenstrual period (23 days)	No-treatment, nonactive lubricants on demand were allowed	CG (*n* = 47)	PE: Vaginal discomfort (VRS) SE: (1) Symptoms (VRS); (2) Vaginal health (VHI mean sum score); (3) Sexual function (FSFI); (4) Sexual distress (FSDS-R).	No severe or serious AEs Mild AEs in 16.7% CG: AEs in 8.5%	↓ Vaginal discomfort ↓ Vaginal symptoms ↑ Vaginal health Sexual function improved ↓ Sexual distress	Level 1	1
Schiavi, M. C., et al. (2019) [34]	Retrospective	Postmenopausal women with VVA. *n* = 172 mean age 60.8 ys	Monurelle Biogel^®^	5 mL once daily for 12 weeks	No CG	No CG	PE: Vaginal health (VHI); SE: (1) Sexual function (FSFI); (2) Sexual distress (FSDS); (3) Urinary symptoms (4) Urogenital distress (UDI-6); (5) Overactive bladder symptoms (OAB-Q) (6) QoL (HRQL)	No significant AEs	↑ Vaginal health Sexual function improved ↓ Sexual distress ↓ Urinary symptoms ↓ Urogenital distress ↓ OAB symptoms ↑ QoL	Level 3	3
Stefani, C., et al. (2014) [35]	Retrospective	Women diagnosed as CIN1. *n* = 256; mean age 37.7 ys	Ginedie^®^ vaginal tablets	twice/week at bedtime for 6 months.	No CG	No CG	ORR to negative histology (Cervical cytology)	NR	75.5% ORR	Level 4	3

OCEBM = Oxford Centre for Evidence Based Medicine; EPHPP = Effective Public Health Practice Project; RCT = Randomized Controlled Trial; ys = years; TG = Treatment Group; CG = Control Group; PE = Primary Endpoint; SE = Secondary Endpoint; VRS = Verbal Rating Scale; VHI = Vaginal Health Index; FSFI = Female Sexual Function Index; FSDS-R = Female Sexual Distress Scale-Revised; AEs = Adverse Events; VVA = Vulvovaginal Atrophy; UDI-6 = Urogenital Distress Index-6 questionnaire; OAB-Q = Overactive Bladder Questionnaire; HRQL = Health Related Quality of Life questionnaire; QoL = Quality of Life; CIN1 = Atypical squamous intraepithelial lesions; NR = Not Reported; ORR = Overall Reduction Rate. Symbols: ↑ = increased; ↓ = decreased.

**Table 3 nutrients-13-02194-t003:** BC as dietary supplement in the sporting population.

Authors	Study Design	Population Number Groups Gender Mean Age	Intervention Matrix	TG size Dosage Frequency Duration	Control Matrix	CG Size Dosage Frequency Duration	Endpoints/Data Collection Tools	Adverse Events	Results	OCEBM	EPHPP
Jones, A. W., et al. (2014) [36]	RCT, DB, PC	Regularly exercising males *n* = 53; mean age 50.5 ys Colostrum (*n* = 25) vs. Placebo (*n* = 28)	BC (Neovite^®^ UK, London)	TG (*n* = 25) 20 g/day 12 weeks.	Isoenergetic/ Isomacronutrient placebo	CG (*n* = 28) 20 g/day 12 weeks.	PE: (1) Incidence of URI; SE: (1) URI days; (2) URI duration (episodes) (3) Immune-system parameters (4) Salivary antimicrobial proprieties (sIgA/AMPs) (5) Salivary microbiome composition	NR	↓ URI incidence; ↓ URI days; ↓ salivary bacterial load; No significant effects on: severity and duration of URI episodes, immune-system parameters, salivary sIgA/AMPs	Level 2	1
Davison G., et al. (2019) [39]	RCT, DB, PC, CB, CO	Recreationally active males *n* = 16; mean age 25.0 ys CO after a week	BC (Neovite^®^ UK, London) during 4.5 h long moderate exercise	TG (*n* = 16): 40 g	Isoenergetic/ Isomacronutrient placebo during 4.5 h long moderate exercise	CG (*n* = 16) 40 g	IGF-1 blood levels	NR	No significant effects on IGF-1 blood levels	Level 2	1
	RCT, DB, PC	Recreationally active males *n* = 20; mean age 28.0 ys	BC (Neovite^®^ UK, London) + training program	TG (*n* = 10) 20 g/day 4 weeks	Isoenergetic/ Isomacronutrient Placebo + training program	CG (*n* = 10) 20 g/day 4 weeks	IGF-1 blood levels	NR	No significant effects on IGF-1 blood levels	Level 3	
	RCT, DB, PC	Recreationally active males *n* = 57; mean age NR Colostrum (*n* = NR) vs. Placebo (*n* = NR) *n* = 4 excluded from the analysis	BC (Neovite^®^ UK, London) + training program	TG (*n* = 25): BC 20 g/day 12 weeks	Isoenergetic/ Isomacronutrient Placebo + training program	CG (*n* = 28) 20 g/day 12 weeks	IGF-1 blood levels	NR	No significant effects on IGF-1 blood levels	Level 3	
Halasa, M., et al. (2017) [40]	RCT, DB, PC	Competitive athletic males *n* = 16; mean age 27.5 ys	Freeze-dried whole BC obtained within 2 h of calf delivery (Genactiv^®^, Poznan, Poland) was packaged in pouches BC 500 mg and desiccated banana 500 mg.	TG (*n* = 8) 1 g/day 20 days	Identical pouches (500 mg of dehydrated whey and 500 mg of desiccated banana) were used as the placebo.	CG (*n* = 8) 1 g/day 20 days	Gut permeability: sugar absorption test, zonulin concentration	No AEs in the TG. Mild AEs in 50% of CG	↓ sugar absorption ↓ zonulin concentration	Level 3	1
Jones, A. W., et al. (2019) [38]	RCT, DB, PC	Recreationally active males *n* = 34; mean age NR Colostum (*n* = 17) vs. Placebo (*n* = 17) *n* = 3 excluded from the analysis	BC + water + training program Day 28: 2 h of 60% maximal aerobic capacity and immune system sensitisation Day 56: elicitation of immunity	TG (*n* = 15) 20 g/day 58 days.	isoenergetic/ isomacronutrient Placebo + training program Day 28: 2 h of 60% maximal aerobic capacity and immune system sensitisation Day 56: elicitation of immunity	CG (*n* = 16) 20 g/day 58 days	PE: Cell mediated response following prolonged exercise (skinfold reactivity) SE: (1) IGF-1 blood levels (2) Immune cell counts (3) Biochemical parameters	NR	↓immune sensitivity decreasing after prolonged exercise No significant effects on in-vivo immune responsiveness, IGF-1 blood levels, Immune cell counts and other biochemical parameters	Level 2	1
Jones, A. W., et al. (2015) [37]	RCT, DB, PC	Recreationally active males *n* = 20, mean age 28.0 ys	BC (Neovite^®^ UK, London)	TG (*n* = 10) 20 g/day 4 weeks	isoenergetic/ isomacronutrient Placebo	CG (*n* = 10) 20 g/day 4 weeks	PE: (1) In-vitro blood neutrophil function: fMLP and PMA (2) Mucosal responses: sIgA and AMP SE: (1) Circulating cells count (2) Biochemical parameters	NR	Beneficial in vitro effects on receptor- dependent (fMLP-stimulated) oxidative burst responses. No in vitro effect on PMA-stimulated oxidative burst, sIgA and AMP. No effects on leukocyte trafficking and other biochemical parameters	Level 2	1
Kotsis, Y., et al. (2019) [42]	RCT, DB, PC	Soccer players *n* = 22; mean age 21.1 ys	Commercial BC 378 Kcal, 67 g protein, 17 g carbohydrates and 4.7 g fat per 100 g Pre and post supplementation LIST exercise program	TG (*n* = 11) 3.2 g/day 6 weeks.	Commercial whey protein 369 Kcal, 90 g protein, 1 g carbohydrates and 0.5 g fat per 100 g Pre and post supplementation LIST exercise program	CG (*n* = 11) 3.2 g/day 6 weeks.	Post-LIST RTD reduction	NR	↓ RTD decline in both groups without significant difference	Level 3	1
Kotsis, Y., et al. (2018) [41]	RCT, DB, PC	Soccer players *n* = 22; mean age NR *n* = 4 excluded from the analysis	Commercial BC 378 Kcal, 67 g protein, 17 g carbohydrates and 4.7 g fat per 100 g Pre and post supplementation LIST exercise program	TG (*n* = 10) 3.2 g/day 6 weeks + 4 days	Commercial whey protein 369 Kcal, 90 g protein, 1 g carbohydrates and 0.5 g fat per 100 g Pre and post supplementation LIST exercise program	CG (*n* = 8) 3.2 g/day 6 weeks + 4 days	EIMD: MIVC, SQJ, CMJ, PMS, biochemical parameters	NR	↑ SQJ, CRP, CK, IL-6 recovery. No significant differences on MIVC, CMJ, PMS and other outcome	Level 2	1
March, D. S., et al. (2018) [43]	RCT, DB, PC, CO	Regularly exercising males *n* = 12; mean age 26 ys Colostrum (*n* = 12) vs. Placebo (*n* = 12) CO after 2 weeks of washout	BC (Neovite^®^ UK, London)	TG (*n* = 12): BC 20 g/day for 14 days Exercise program 70% aerobic capacity for 1 h		CG (*n* = 12): Isoenergetic/Isomacronutrient placebo for 14 days Exercise program 70% aerobic capacity for 1 h	PE: Exercise-induced intestinal cell damage (I-FABP) SE: (1) Bacterial translocation (plasmatic bacterial DNA) (2) Other physical parameters	NR	↓ I-FABP plasma concentration after exercise No effects Bacterial DNA plasmatic concentration No significant differences on other outcome	Level 2	1
Morrison, S. A., et al. (2014) [45]	RCT, DB, PC, CO	Trained and Untrained males *n* = 28 (14 trained, 14 untrained) Colostrum (*n* = 14) vs. Placebo (*n* = 14) *n* = 13 (6 untrained, 7 trained) lost and excluded from the analysis CO time NR	BC (Hokitika, New Zealand) protein, 58.2% m/m; fat, 1.4% m/m; lactose, 29.3% m/m; and IgG, 15.3%. Before 90 min multi-mode exercise session	TG (*n* = 7 trained, *n* = 8 untrained): BC 1.7 g/kg/day for 7 days	Corn flour placebo	CG (*n* = 7 trained, *n* = 8 untrained): corn flour placebo for 7 days before 90 min multi-mode exercise session	PE: GI permeability (Double sugar model, I-FABP) SE: (1) cytokine level and other blood parameters (2) thermal and cardiovascular measures (3) Other parameters	No AEs	↑ I-FABP in trained group No significant differences on other outcome	Level 3	1
Morrison, S. A., et al. (2013) [44]	RCT, DB, PC, CO	Healthy males *n* = 15 (7 highly-fit, 8 moderately-fit); mean age 22 ys	BC (Hokitika, New Zealand) protein, 58.2% m/m; fat, 1.4% m/m; lactose, 29.3% m/m; and IgG, 15.3%. Before 90 min multi-mode exercise session	TG (*n* = 7 highly fit, *n* = 8 moderately-fit) 1.7 g/kg/day 7 days	Corn flour placebo	CG (*n* = 7 highly fit, *n* = 8 moderately fit) 1.77 g/kg/day 7 days	PE: (1) BBB permeability (S100ß protein, cerebral oxygenation) (2) Cognitive function (Stroop test and perceptions) SE: (1) thermal and cardiovascular measures (2) Other parameters	NR	No effects on BBB, cognitive and physical performance	Level 3	1
Shing, C. M., et al. (2013) [46]	RCT, DB, PC Pilot study	Highly-trained males *n* = 10; mean age NR CPC (*n* = 4) vs. WPC placebo (*n* = 6)	Intact^®^ bovine CPC (Numico Research Australia Pty Ltd., South Australia) before and during a 5 days cycling race	TG (*n* = 4) 10 g/day for 8 weeks + 5 days	Whey protein concentrate (Alacen^®^ 80’’ Fonterra Co-op Group Limited, Auckland, New Zealand) before and during a 5 days cycling race	CG (*n* = 6) 10 g/day for 8 weeks + 5 days	PE: hormonal (salivary hormones level), immune (salivary IgA) and autonomic (parasympathetic indices of HRV) response SE: Mood profile (POMS)	NR	↑ testosterone concentration maintenance ↑ cortisol concentration before the race ↑ parasympathetic indices of HRV No significant differences on cortisol concentration during race, POMS and salivary IgA concentration	Level 3	1

OCEBM = Oxford Centre for Evidence Based Medicine; EPHPP = Effective Public Health Practice Project, RCT = Randomized Controlled Trial; DB = Double Blind; PC = Placebo Controlled; CB = Conterbalanced; CO = Crossover; ys = years; TG = Treatment Group; BC = Bovine Colostrum; CG = Control Group; PE = Primary Endpoint; URI = Upper Respiratory Illness; IGF-1 = Insulin like Growth Factor 1; sIgA = salivary Immunoglobulin A; AMP = Anti-Microbial Peptides; SE = Secondary Endpoint; NR = Not Reported; AEs = Adverse Events; fMLP = formylmethionyl-leucyl phenylalanine; PMA = phorbol-12-myristate-13-acetate; LIST = Loughborough Intermittent Shuttle Test; RTD = Rate of Torque; EIMD = Exercise Induced Muscle Damage; MIVC = Maximum Isometric Voluntary Contraction; SQJ = Squat Jump; CMJ = Countermovement Jump; PMS = Perceived Muscle Soreness; CRP = C-Reactive Protein; CK = Creatin Kinase; IL-6 = Interleukin-6; I-FABP = Intestinal Fatty Acid Binding Protein; BBB = Blood Brain Barrier; CPC = Colostrum Protein Concentrate; WPC = Whey Protein Concentrate; HRV = Hearth Rate Variability; POMS = Profile Of Mood States questionnaire. Symbols: ↑ = increased; ↓ = decreased.

**Table 4 nutrients-13-02194-t004:** BC as dietary supplement in the pediatric and preterm infants.

Authors	Study Design	Population Number Groups Gender Mean Age	Intervention Matrix	TG size Dosage Frequency Duration	Control Matrix	CG Size Dosage Frequency Duration	Endpoints/Data Collection Tools	Adverse Events	Results	OCEBM	EPHPP
Aunsholt, L., et al. (2014) [47]	RCT, DB, PC, CO, PS	Children with SBS *n* = 9 (4 females, 5 males); median age 39 months CO after 4 weeks washout	“Fresh” BC from 15 different cows (Danish Holstein) within the first 24 h after calving	TG (*n* = 9) 20% of BFR	Mixed milk, cream, and whey protein	CG (*n* = 9) 20% of BFR	PE: Nutrients and Fluid balances SE: Anthropometric, knemometry, biological parameters	NR	No significant differences in intestinal energy and wet weight absorption, knemometry, IGF-1, IGF-BP3 levels	Level 2	1
Balachandran, B., et al. (2016) [48]	RCT, DB, PC, PS	VLBW Neonates *n* = 86; chronological age < 96 h Colostrum (*n* = 43) vs. Placebo (*n* = 43)	BC Pedimmune^®^ (Mumbai, India)	TG (*n* = 43) 1.2–2.0 g/dose + feeding 4 times a day 21 days	Equal dose placebo (not specified)	CG (*n* = 43) 1.2–2.0 g/die + feeding 4 times a day 21 days	PE: NECSE: sepsis, mortality and stool interleukin-6 (IL-6) levels	No AEs	No significant differences in NEC, sepsis and mortality. ↑IL-6 and radiological features of NEC in TG	Level 2	1
Barakat et al. (2019) [49]	RCT, DB, PC	Pediatrics with acute diarrhea *n* = 160; aged 6 months to 2 ys	ImmuGuard^®^, sachets (London, England)	TG (*n* = 80) 3 g/sachet + standard therapy	Equal dose pla-cebo (not speci-fied)	CG (*n* = 80) + standard therapy	PE: Diarrhea frequency and duration, vomiting duration SE: fever duration and Vesikari scoring	NR	↓ Diarrhea and vomit frequency and duration ↓ Vesikari Scoring after 48 h	Level 2	1
Meinich Juhl, S., et al. (2018) [50]	RCT, OL, PS	NPI *n* = 40; gestational age 27–32 weeks Country stratification (China-Denmark)	Unmodified intact BC powder (ColoDan^®^; Gesten, Denmark) as supplement of MM, DM or IF	TG (*n* = 21) max 4.5 g/kg/day 10–14 days	Standard feeding with MM, DM or IF	CG (*n* = 19) 10–14 days	PE: Tolerability and safety SE: nutritional outcomes	1 death for NEC not related to BC intake Late onset sepsis (2 TG, 1 CG), pulmonary dysplasia (1 TG), ROP (2 TG), metabolic acidosis (2 TG) No significant differences	↑ protein intake in TG (China group) ↑ in TG Temporary elevation in plasmatic tyrosine levels on day 7 No significant differences on dietary intolerances and other outcomes	Level 3	2
Patıroğlu, T. and M. Kondolot (2013) [51]	RCT, DB, PC	IgA deficient paediatrics with viral URI *n* = 31; median age 8.5 ys; *n* = 18 males and 13 females Colostrum (*n* = 16) vs. Placebo (*n* = 15)	BC sucking tablet that contains 14 mg of colostrum and 2.2 mg of lysozyme (Igazym^®^; Vejle, Denmark)	TG (*n* = 16) 3 times a day 1 week	placebo sucking tablets	CG (*n* = 15) 3 times a day for 1 week	PE: sIgA SE: Infection severity	No AEs 1 patient included 2 times and 1 included 3 time for different infections	No significant differences in sIgA secretion ↓infection severity score in BC after 1 week	Level 2	1
Rathe, M., et al. (2020) [52]	RCT, DB, PC Multicentre study	Pediatrics with ALL *n* = 62, aged 1–18 ys *n* = 32 males and 30 females Colostrum (*n* = 30) vs. Placebo (*n* = 32)	Intact, spray-dried BC powder (Gesten, Denmark^®^).	TG (*n* = 30) 0.5–1 g/kg/day 4 weeks	Isocaloric placebo. whole-milk powder enriched with whey protein isolate powder	CG (*n* = 32) 4 weeks	PE: fever level and duration.SE: CRP levels, neutrophil count, bacteraemia or fungaemia episodes, treatment delay, mucositis severity, PROs on chemotherapy toxicity, compliance	No AEs	↓ Peak severity of oral mucositis No significant differences on PE and other SE, low compliance reported	Level 2	1
Saad, K., et al. (2016) [53]	Cohort Prospective Multicentric	Children with recurrent acute URI or diarrhea due to infection *n* = 160; aged 1–6 ys. *n* = 81 males and 79 females	ImmuGuard^®^, sachets (London, England)	TG *n* = 160 3 g/sachet 4 weeks.	No CG	No CG	URI or diarrhea episodes and frequency of hospitalizations Follow up period 24 weeks	Mild transient AEs reported in 12 patients: Skin rush (9), itching (1), and diarrhea (2). 6 patients discontinued the BC treatment.	↓ Infection episodes at 2 and 6 months ↓ Hospitalizations	Level 3	3

OCEBM = Oxford Centre for Evidence Based Medicine; EPHPP = Effective Public Health Practice Project, RCT = Randomized Controlled Trial; DB = Double Blind; PC = Placebo Controlled; CO = Crossover; PS = Pilot Study; SBS = Short bowel syndrome; ys = years; TG = Treatment Group; BC = Bovine Colostrum; BFR = Basal Fluid Requirement; CG = Control Group; PE = Primary Endpoint; IGF-1 = Insulin like Growth Factor 1; IGF-BP3 = Insulin like Growth Factor Binding Protein 3; VLBW = Very Low Birth Weight; NEC = Necrotizing Enterocolitis; OL = Open Label; NPI = Newborn Preterm Infants; MM = Mother Milk; DM = Donor Milk, IF = Infant Formulae; ROP = Retinopathy of Prematurity; URI = Upper Respiratory Illness; sIgA = salivary Immunoglobulin A; ALL = Acute Limphoblastic Leukemia; CRP = C-Reactive Protein. Symbols: ↑ = increased; ↓ = decreased.

**Table 5 nutrients-13-02194-t005:** BC as dietary supplement in healthy older adults.

Authors	Study Design	Population Number Groups Gender Mean Age	Intervention Matrix	TG Size Dosage Frequency Duration	Control Matrix	CG Size Dosage Frequency Duration	Endpoints/Data Collection Tools	Adverse Events	Results	OCEBM	EPHPP
Duff, W. R., et al. (2014) [54]	RCT, DB, PC	Older adults *n* = 40; mean age 59 ys *n* = 15 males and 25 females	spray-dried BC (Eterna Gold^®^, Saskatoon, Canada) + resistance training program	TG (*n* = 19) 60 g/day of BC 8 weeks	whey placebo (Cereal^®^, Illinois, US) + resistance training program	CG (*n* = 21): 38 g/day WP complex (60 g total) for 8 weeks	PE: Body composition and strength SE: Muscle thickness, serum assessment, bone resorption, cognitive function	Mild-moderate GI AEs was reported by 5 participants (2 in TG, 3 in CG)	↑ leg press strength ↓ bone resorption No differences in body composition, muscle thickness and other outcomes	Level 2	1

OCEBM = Oxford Centre for Evidence Based Medicine; EPHPP = Effective Public Health Practice Project, RCT = Randomized Controlled Trial; DB = Double Blind; PC = Placebo Controlled; ys = years; TG = Treatment Group; BC = Bovine Colostrum; CG = Control Group; WP = Whey Protein; PE = Primary Endpoint; SE = Secondary Endpoint; GI = Gastrointestinal; AEs = Adverse Events. Symbols: ↑ = increased; ↓ = decreased.

**Table 6 nutrients-13-02194-t006:** BC administration in critically ill patients.

Authors	Study Design	Population Number Groups Gender Mean Age	Intervention Matrix	TG Size Dosage Frequency Duration	Control Matrix	CG Size Dosage Frequency Duration	Endpoints/Data Collection Tools	Adverse Events	Results	OCEBM	EPHPP
Eslamian, G., et al. (2019) [55]	RCT, DB, PC	ICU patients *n* = 70; mean age 62 ys Colostrum (*n* = 35) vs. Placebo (*n* = 35) *n* = 8 not included in the analysis	Neovite ^®^ (London, UK).	TG (*n* = 32): BC 20 g/day + enteral feeding for 10 days	ENTERA Meal^®^; (Tehran, Iran)	CG (*n* = 30): Isocaloric maltodextrin + enteral feeding for 10 days	PE: Intestinal permeability (plasmatic endotoxin and zonulin) SE: mortality, LOS, GI complications	No AEs	↓ Plasmatic endotoxin and zonulin concentrations at day 10. ↓ Diarrhea No significant differences in other comparisons	Level 2	1

OCEBM = Oxford Centre for Evidence Based Medicine; EPHPP = Effective Public Health Practice Project, RCT = Randomized Controlled Trial; DB = Double Blind; PC = Placebo Controlled; ICU = Intensive Care Unit; ys = years; TG = Treatment Group; BC = Bovine Colostrum; CG = Control Group; WP = Whey Protein; PE = Primary Endpoint; SE = Secondary Endpoint; LOS = Length of Stay; GI = Gastrointestinal; AEs = Adverse Events. Symbol: ↓ = decreased.

**Table 7 nutrients-13-02194-t007:** Abstracts on BC clinical applications.

Authors	Study Design	Population Number Groups Gender Mean Age	Interventions Dosage Frequency Duration	Control Dosage Frequency Duration	Endpoints Data Collection Tools	Adverse Events	Results	OCEBM
Caysido et al. (2017) [56]	RCT, DB, PC	Pediatrics undergoing chemotherapy for ALL *n* = 21; aged 6 months to 18 ys	TG (*n* = NR): BC twice a day for a week from the first day of chemotherapy	CG (*n* = NR) Placebo twice a day for a week from the first day of chemotherapy	PE: neutropenia (CBC, ANC)	No AEs	↑ ANC, WBC and PLT blood levels	Level 5
Barakat, S., et al. (2019)b [57]	RCT, DB, PC	Pediatrics with acute diarrhea *n* = 160; aged 6 months to 2 ys	TG (*n* = 80) BC 3 g/day for 1 week	CG (*n* = 80) placebo for 1 week	PE: *n*° of patients with diarrhea after 72 h	NR	Diarrhea stopped in 65% of TG vs. 95% of CG after 72 h	Level 5
Donowitz, J., et al. (2019) [58]	RCT, PC	Infants (income country) *n* = NR; 6 to 9 months	TG (*n* = NR) 7 g of PTM202 twice a day 30 days	CG (*n* = NR) micronutrient sprinkles twice a day 30 days	PE: EED (fecal MPO, fecal Reg 1B, serum CRP, serum sCD14, and L:M	NR	↓ fecal MPO and reg1B No significant differences in other parameters	Level 5
Oloroso-Chavez, K., et al. (2017) [59]	RCT Subgroup analysis	Pediatrics with respiratory allergies *n* = 38; aged 7 to 18 ys	TG (*n* = 19) BC 1000 mg day 3 months.	CG (*n* = 19) Placebo 1000 mg day 3 months	PE: Symptoms improvement (TNSS, ACT, CASI and pulmonary function test)	NR	↓ nasal congestion (TNSS) and lung function in monosensitized subjects ↑ ACT and CASI scores in polysensitized subjects	Level 5

OCEBM = Oxford Center for Evidence Based Medicine; RCT = Randomized Controlled trial; DB = Double Blind, PC = Placebo Controlled, ALL = Acute Lymphoblastic Leukemia; ys = years; TG = Treatment Group; NR = Not Reported; BC = Bovine Colostrum; CG = Control Group; PE = Primary Endpoint; CBC = Complete Blood Count; ANC = Absolute Neutrophil Count; AEs = Adverse Events; WBC = White Blood Cells; PLT = Platelets; PTM202 = Nutritional Supplement with Bovine Colostrum; EED = Environmental Enteric Dysfunction; MPO = Myeloperoxidase; Reg 1B = Regenerating Gene 1B; sCD14 = plasma soluble CD14; CRP = C-Reactive Protein; L:M = Lactulose Mannitol ratio; TNSS = Total Nasal Symptoms Score; ACT = Asthma Control Test; CASI = Composite Asthma Severity Index. Symbols: ↑ = increased; ↓ = decreased.

**Table 8 nutrients-13-02194-t008:** Systematic reviews on BC clinical applications.

Authors	Study Design	n of Paper Included Heterogeneity n of Participants	Bovine Colostrum Effects Measured	Evidence	Adverse Events	OCEBM	AMSTAR 2
Rathe, M., et al. (2014) [21]	Systematic review	49 record covering 51 studies High heterogeneity: settings, methodologies, treatment/placebos preparations and dosages, population, diseases, endpoints and outcomes 2326 participants	- NSAID and surgery induced GI toxicity (2 RCT—*n* = 122 participants) - HIV associated diarrhea and immunosuppression (1 CSS, 3 OBS, 1 RT—*n* = 182 participants) - Sports nutrition and exercise (12 RCT, 1 NRT, 1 OBS—*n* = 370 participants) - Immune functions in sport and exercise (9 RCT; 1 NRT—*n* = 244 participants) - Infection and immune responses (10 RCT, 1 OBS—*n* = 1090 participants) - SBS (2 RCT—*n* = 21 participants) - Growth and metabolic - Disorders (1 RCT, 1 OBS—*n* = 138 participants) - Juvenile idiopathic arthritis (1 RCT—*n* = 30 participants) - Chronic pain syndrome and irritable bowel syndrome (1 RCT, 1 CSS—*n* = 114 participants)	- Evidence suggests that BC protect GI tract from NSAID (short period) induced injuries - Evidence suggests that BC may reduce microbial translocation across the gut mucosa in patients undergoing abdominal surgery - Evidence suggests that BC can effectively ameliorate HIV-associated diarrhea - Contradictory evidence are shown on the effects of bovine colostrum on sports performance, body composition, and nutrient absorption - Not conclusive evidence are available on the effects of colostrum on immunity - Evidence suggests that BC triggers immunological events followed by systemic ones - No evidence of a marked effect on intestinal function has been documented in SBS patient - Inconclusive evidence of benefit of BC on thrive failures and on metabolic control of type II diabetes - Evidence of no effect of BC supplementation on clinical outcomes - Evidence of no significant effect of BC on irritable bowel syndrome	No serious AEs Mild/moderate AEs were reported: unpleasant taste, nausea, flatulence, diarrhea, skin rash, and unspecified abdominal discomfort. Nine studies reported an absence of side effects.	Level 1	HIGH

OCEBM = Oxford Centre for Evidence Based Medicine; AMSTAR 2 = A Measurement Tool to Assess Systematic Reviews 2; NSAID = Nonsteroidal anti-inflammatory drug; GI = Gastrointestinal; RCT = Randomized Controlled Trial; HIV = Human Immunodeficiency Virus; CSS = Cross Sectional Study; OBS = Observational Study; RT = Randomized Trial; NRT = No Randomized Trial; SBS = Short Bowel Syndrome; BC = Bovine Colostrum; AEs = Adverse Events.

**Table 9 nutrients-13-02194-t009:** CCRBT bias assessment.

	Random Sequence Generation (Selection Bias)	Allocation Concealment (Selection Bias)	Blinding of Participants and Personnel (Performance Bias)	Blinding of Outcome Assessment (Detection Bias)	Incomplete Outcome Data (Attrition Bias)	Selective Reporting (Reporting Bias)	Other Bias
Nappi, R. E., et al. (2016)	+	+	-	?	+	?	+
Jones, A. W., et al. (2014)	?	?	+	+	+	+	+
Davison G., et al. (2019)	?	?	+	+	+	?	+
Halasa, M., et al. (2017)	+	+	+	+	+	?	?
Jones, A. W., et al. (2019)	+	+	+	+	+	?	?
Jones A.W., et al. (2015)	?	?	+	+	+	+	+
Kotsis, Y., et al. (2019)	+	+	+	+	?	?	+
Kotsis, Y., et al. (2018)	+	+	+	+	+	+	?
March, D. S., et al. (2018)	+	+	+	+	+	+	+
Morrison S. A., et al. (2014)	?	?	+	+	-	?	?
Morrison, S. A., et al. (2013)	?	?	+	+	+	?	-
Shing, C. M., et al. (2013)	+	?	+	?	+	?	+
Aunsholt, L., et al. (2014)	+	?	+	+	+	+	+
Balachandran, B., et al. (2016)	+	+	+	?	+	?	?
Barakat et al. (2019)	+	+	+	+	+	+	+
Meinich Juhl, S., et al. (2018)	+	+	-	-	+	?	?
Patıroğlu, T. and M. Kondolot (2013)	+	?	+	?	+	+	+
Rathe, M., et al. (2020)	+	+	+	+	+	?	+
Duff, W. R., et al. (2014)	+	+	+	+	+	+	+
Eslamian, G., et al. (2019)	+	+	+	+	+	+	+

## Data Availability

No new data were created or analyzed in this study. Data sharing is not applicable to this article.

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
