# Peer review of "Bovine Colostrum Applications in Sick and Healthy People: A Systematic Review"

_nutrients, 2021, doi:10.3390/nu13072194_

Round 1

Reviewer 1 Report

Meta-analysis is a systematic review of a focused topic in the literature that provides a quantitative estimate for the effect of a treatment intervention or exposure. Differing from a systematic review, unsystematic narrative review tends to be descriptive, in which the authors select frequently articles based on their point of view which leads to its poor quality. Systematic review/meta-analysis steps include development of research question and its validation, forming criteria, search strategy, searching databases, importing all results to a library and exporting to an excel sheet, protocol writing and registration, title and abstract screening, full-text screening, manual searching, extracting data and assessing its quality, data checking, conducting statistical analysis, double data checking, manuscript writing, revising, and submitting to a journal. The method used to pool the studies appears to have taken no account of the size of the studies. It is clear that the studies included in the effect size analysis, were able to contribute a measure of precision, but the effect sizes used are adjusted only for inter-patient variation and not sample size.Therefore, size should be taken into account, because conclusions may be misleading.

Author Response

The authors thanks the reviewers for their very useful comments and suggestions. We though to satisfy them revisiting the wole paper structure. The Results chapter has been expanded and divided in sub-chapters. The Discussion chapter has been revisited avoiding redundancies. The tables has been placed in horizontal format in order to favour their readability. A clean version of the paper was uploaded to facilitate its reading by the reviewers. PLEASE, LOOK AT THE ATTACHED FILE FOR DETAILS.

Reviewer 2 Report

I’m a bit torn on this paper. The title promises a systematic review, but I am unconvinced if it truly does. While some aspects of a systematic review are clearly followed, I miss the clearly defined research question for this purpose. The topic that the authors aim to cover is so broad and so scattered that one would have to wonder whether a systematic review is possible If I read the paper, I feel that it has become more an overview than a review, with hardly ever direct linkages between papers. As such, I still feel I would really need to go back to the pretty much all the original papers.

My other issue is that the authors fail to give perspective on how the colostrum was used. In most cases, this will of course be as a supplement that has undergone preprocessing and is applied in a matrix. That, however, is very unclear from the paper and I think is needed to put any perspective on any outcomes.

Overall, I suppose the conclusions section kind of summarizes my feelings after reading the paper. A good attempt, but the end result is a bit disappointing. I don’t think this is due to the authors efforts, but rather due to the area covered. I will leave it up to the editor to decide how to deal with this.

Author Response

(The authors gave the same response as above.)

Reviewer 3 Report

I have reviewed the manuscript and recommend minor revisions. The subject is of interest for readers of Nutrients while the aim of the study is well designed. Even the conclusions of the systematic review points to further studies, avoiding confounding factors and biases in design to obtain in the future evidence to support the application of BC in some diseases or un-healthy conditions.

I suggest the authors to rephrase the statement in lines 74-75.

In Table 1, what is the meaning of the term dramatic effect?

Author Response

The authors thanks the reviewers for their very useful comments and suggestions. We though to satisfy them revisiting the wole paper structure. The Results chapter has been expanded and divided in sub-chapters. The Discussion chapter has been revisited avoiding redundacies. The tables has been placed in horizontal format in order to favour their readability. A clean version of the paper was uploaded to facilitate its reading by the reviewers. PLEASE, LOOK AT THE ATTACHED FILE FOR DETAILS

Round 2

Reviewer 1 Report

Thank you for taking into account the comments.

Reviewer 2 Report

I'm pleased with the revisions made by the authors